# Simultaneous Effect of Diameter and Concentration of Multi-Walled Carbon Nanotubes on Mechanical and Electrical Properties of Cement Mortars: With and without Biosilica

**DOI:** 10.3390/nano14151271

**Published:** 2024-07-29

**Authors:** Suren A. Malumyan, Nelli G. Muradyan, Marine A. Kalantaryan, Avetik A. Arzumanyan, Yeghvard Melikyan, David Laroze, Manuk G. Barseghyan

**Affiliations:** 1Faculty of Construction, National University of Architecture and Construction of Armenia, 105 Teryan Street, Yerevan 0009, Armenia; surmalumyan@gmail.com (S.A.M.); nmuradyan@nuaca.am (N.G.M.); marakalantaryan60@gmail.com (M.A.K.); av.arzumanyan@nuaca.am (A.A.A.); 2A.B. Nalbandyan Institute of Chemical Physics, NAS RA, 5/2P. Sevak Street, Yerevan 0014, Armenia; yeghvard.melikyan1@edu.isec.am; 3Instituto de Alta Investigación, Universidad de Tarapacá, Casilla 7D, Arica 1000000, Chile; dlarozen@academicos.uta.cl

**Keywords:** carbon nanotube, biosilica, mortar, compressive strength, electrical resistivity, piezoresistivity

## Abstract

In this work, the effect of multi-walled carbon nanotubes (MWCNT1, MWCNT2, and MWCNT3) with different outer diameters and specific surface areas on the mechanical and electrical properties of cement mortar have been investigated. Various concentrations of MWCNTs were used (0.05, 0.10, and 0.15%), the effective dispersion of which was carried out by an Ultrasonic machine (for 40 min with 160 W power and a 24 kHz frequency) using a surfactant. Composites have been processed with a biosilica content of 10% by weight of cement and without it. Compressive strength tests were carried out on days 7 and 28 of curing. The 7-day compressive strength of samples prepared without biosilica increased compared to the result of the control sample (6.4% for MWCNT1, 7.4% for MWCNT2, and 10.8% for MWCNT3), as did those using biosilica (6.7% in the case of MWCNT1, 29.2% for MWCNT2, and 2.1% for MWCNT3). Compressive strength tests of 28-day specimens yielded the following results: 21.7% for MWCNT1, 3.8% for MWCNT2, and 4.2% for MWCNT3 in the absence of biosilica and 8.5%, 12.6%, and 6.3% with biosilica, respectively. The maximum increase in compressive strength was observed in the composites treated with a 0.1% MWCNT concentration, while in the case of 0.05 and 0.15% concentrations, the compressive strengths were relatively low. The MWCNT-reinforced cement matrix obtained electrical properties due to the high electrical conductivity of these particles. The effect of MWCNT concentrations of 0.05, 0.10, and 0.15 wt% on the electrical properties of cement mortar, especially the bulk electrical resistivity and piezoresistive characteristics of cement mortar, was studied in this work. At a concentration of 0.05%, the lowest value of resistivity was obtained, and then it started to increase. The obtained results show that all investigated specimens have piezoresistive properties and that the measurements led to a deviation in fractional change in resistivity.

## 1. Introduction

Cement-based materials have become one of the most commonly used materials in civil engineering. Compressive and flexural strengths, elastic modulus, and fracture toughness are vital properties of structural materials. Improving these properties of cement-based materials, mainly compressive strength and toughness, is challenging. For this purpose, some studies have started to investigate the utilization of nanomaterials, such as nanofibers [1,2], nanosilica [3,4], and carbon nanotubes CNTs [5,6,7] to improve the characteristics of cement-based materials. It is well known that incorporating CNTs into the matrices of mortars and concrete based on Portland cement can enhance their durability and mechanical properties. CNTs are known for their exceptional strength, stiffness, and electrical conductivity, which can significantly improve the performance of cement-based materials. CNTs can significantly improve cement composites’ compressive, tensile, and flexural strength due to their high aspect ratio and superior mechanical properties. CNTs help bridge micro-cracks and delay their propagation, enhancing the toughness and ductility of the composite. CNTs fill the pores and refine the microstructure of the cement matrix, reducing permeability and enhancing resistance to water and chemical ingress. The presence of CNTs can help resist the growth of micro-cracks, thereby enhancing the durability and longevity of the composite. CNTs impart electrical conductivity to cement composites, enabling applications in the self-sensing and monitoring of the structural health of concrete. Ultrasonication is commonly used to disperse CNTs in a liquid medium before mixing them into the cement paste. Surfactants or dispersing agents are often added to stabilize the dispersion. CNTs act as reinforcing agents in cement composites, providing high strength, elasticity, and isotropic properties while reducing the porosity of the mixtures [8,9,10,11,12].

It is important to note that the quality of composite materials that contain nanomaterials is dependent on how well the CNTs are dispersed within the forming matrix. The better the dispersion and more uniform distribution of CNTs within the cement skeleton, the greater the control over the properties of the resulting composite materials [13,14]. The dispersion of nanomaterials in water used for mixing mortar or concrete is the first step in achieving a uniform distribution of nanomaterials in the cement matrix. The incorporation of carbon nanotubes (CNTs) and silica fume into cement paste involves specific procedures to ensure proper dispersion and enhance the composite’s mechanical and durability properties. Developing aqueous dispersions with optimal concentrations of nanomaterials will make it easier to produce cementitious nanocomposites on a larger scale, with an optimal volume fraction of nanomaterials [15,16]. The pozzolanic reaction of silica fume with calcium hydroxide (CH) produces additional C-S-H, improving the density and homogeneity of the cement paste. CNTs impart electrical conductivity to the cement paste, enabling self-sensing capabilities for monitoring structural health. In large-scale industrial concrete production, a significant amount of water is required. When producing concrete nanocomposites, it is beneficial to minimize the amount of water used in dispersing nanomaterials. It is important to note that nanomaterials tend to stick together due to van der Waals interactions caused by their large specific surface areas. The dispersion of nanomaterials in the aqueous environment of cementitious materials will benefit from modifications that make their surface more hydrophilic. These modifications should also enhance the interaction of nanomaterials with cement hydrates [17,18]. The goal of changing cement composite properties at the nanoscale derives from the atomic structure of their primary hydration products, calcium hydroxide (CH) and calcium silicate hydrate (C-S-H). SEM observation identifies the CH gel as solid sheets. The crystallization process of calcium silicate hydrate (C-S-H) is likely to follow a nonclassical crystallization pathway. The structure of C-S-H varies greatly and depends on the raw materials, environment, and reaction stage. It has been reported that more than 40 types of stable C-S-H phases can be determined in hydrated cementitious materials. Although the typical C-S-H gel in concrete is amorphous, the local structure is short-range-ordered in the scale of 3–5 nm. The regular atomic arrangement of C-S-H was considered to be tobermorite-like and jennite-like crystals [5,6].

Besides the advantages of improving almost all mechanical properties, the addition of MWCNTs in the cementitious matrix can result in the tailoring of the electrical properties of the nanocomposites. Limited studies have been carried out on the piezoresistive behavior and sensing ability of cementitious nanocomposites embedded with MWCNTs [19].

Piezoresistivity is the effect of change in the electrical resistivity induced by the deformation of materials [20]. Luo et al. [21] used cured MWCNT-reinforced cement-based composites with 0.1 and 0.5 wt% MWCNTs and measured the electrical resistance under cyclic loading. The results revealed good piezoresistivity and strain sensitivity for both samples, though the trendline of fractional change in resistivity presented better stability for amounts of 0.5 wt%. Han et al. [22] examined cement nanocomposites with 0.05, 0.1, and 1.0 wt% of cement MWCNTs. The experimental results indicated that the piezoresistive sensitivity of the composites first increased and then decreased with the increase in the MWCNT content, concluding that the composite reinforced with 0.1 wt% of MWCNTs presented better sensing properties. Azhari and Banthia [23] investigated the piezoresistivity response of cement-based composites using two types of cement-based sensors, one with carbon fibers (CFs) alone and the other carrying a hybrid of both CFs and CNTs. The authors measured the electrical resistance of the composites using the AC measurement method under cyclic or monotonic compressive loading up to failure. They concluded that the hybrid sensors, containing a combination of CFs and CNTs, provide a better-quality signal, improved reliability, and increased sensitivity over sensors carrying CFs alone. Recently, Kim et al. [24] examined the effect of the water-to-binder ratio on piezoresistivity by fabricating and testing mortar nanocomposites with 0.4, 0.5, and 0.6 water/binder ratios, reinforced with CNTs at amounts of 0.1, 0.3, and 0.5 wt% of cement. The experimental results indicated that the stability of piezoresistivity under cyclic loading and their time-based sensitivity can be improved by decreasing the water/binder ratio of the cement composites. In addition, the variation of piezoresistivity induced by the moisture content can be decreased by low water/binder ratios.

Microsilica [5] and nanosilica [3,4] have been used to produce a variety of cement mortars. All the mentioned materials were combined to change the mortars’ characteristics, thus improving the mortars’ mechanical qualities. The cement composites containing nanosilica were found to accelerate C_3_S hydration and the formation of gel C-S-H. Furthermore, the viscosity of the composite was modified, resulting in an improvement in the stiffness of the cement matrix. Biosilica is also a form of silica-containing material derived from biological sources, such as diatoms [25,26,27]. It typically has a highly porous structure and a high surface area, making it an excellent material for reinforcing cementitious materials. This reinforcement helps to distribute applied stresses more evenly throughout the material, reducing the likelihood of crack propagation and increasing the overall strength and toughness of the cement [27,28,29].

As concrete structures are subjected to diverse loading conditions, they are prone to developing cracks and damage. Consequently, the importance of damage detection and crack monitoring processes significantly rises for ensuring the structural integrity of these constructions. Various techniques are employed for Structural Health Monitoring (SHM), including accelerometers, fiber optics, and acoustic emission sensors [28,30]. The construction industry extensively utilizes bulk and surface resistivity methods for tasks like evaluating steel corrosion, identifying cracks in concrete, and assessing the effects of coarse aggregates and various cement types on concrete’s electrical resistivity. In addition to their mechanical properties, CNTs integrated into cement-based matrices enhance the electrical characteristics due to the high electrical conductivity of carbon nanoparticles. This makes them suitable as self-sensing materials. Using the electrical properties of CNTs enables SHM without the need for additional sensors or tools. The electrical method stands out as one of the top non-destructive techniques for evaluating structural damage and the conditions of concrete structures. This method involves measuring the electrical resistance of the structural material, which changes in response to strain and damage. Electrical-resistance-based strain/damage self-sensing in cement-based materials has been in existence for over 30 years, and the field has significantly advanced in that time [28,29,30,31,32,33,34].

The main goal of this work lies in exploring how Multi-Walled Carbon Nanotubes (MWCNTs) with varying outer diameters and concentrations affect the mechanical and electrical properties of cement mortar with and without biosilica with 7- and 28-day curing periods.

## 2. Materials and Methods

### 2.1. Materials

In this study, M500 ordinary Portland cement (52.5 class) produced by HRAZDAN CEMENT CORPORATION (LLC) (Hrazdan, Armenia) was used as a binder for preparing the cement mortar [35]. The physical and mechanical characteristics and chemical composition were determined using standard methods [36,37,38] and are presented in Table 1.

For the preparation of cement mortar, river sand was used as the fine aggregate, and the average data of its physical characteristics are presented in Table 2 [39].

Biosilica is a form of silica derived from biological sources, such as diatoms or other silica-secreting organisms. It typically has a highly porous structure and a high surface area, making it an excellent candidate for reinforcing cementitious materials. When biosilica is incorporated into cement, its porous structure can act as micro-reinforcements within the cement matrix. Biosilica particles can also improve the hydration kinetics of cement, promoting the formation of hydration products like calcium silicate hydrate (C-S-H) gel. This can lead to denser and more uniform microstructures, further enhancing the mechanical properties of the cement [40]. It contains active silica, which can react with Ca(OH)_2_ at room temperature. Amorphous silica modifies lime into calcium silicate hydrate. The durability increases and the capillary porosity decreases as the capillary reaction products thicken the structure. This contributes to a denser and less permeable cement structure. The natural origin and developed technology provide fixed physicochemical properties and high additive activity, 1.5 times higher than microsilica [25,30,41,42].

The technical characteristics of the biosilica used, sourced from the “EFFECT GROUP” in Yerevan, Armenia, are presented in Table 3.

Chemical compounds of biosilica were characterized by Fourier transform infrared spectroscopy–attenuated total reflectance (Spectrum Two, PerkinElmer, Waltham, MA, USA) in a typical range of 4000 to 400 cm^−1^. Particle size analysis of the biosilica was implemented using the dynamic light scattering (DLS) technique (Litesizer 500, Anton Paar, Graz, Austria).

The FTIR-ATR spectrum of biosilica is presented in Figure 1. As we can see, the dominant peak at 1068 cm^−1^ indicates a combination of asymmetric stretching bonds of silica oxygen (Si–O–Si), the peak at 799 cm^−1^ is due to silica oxygen bond symmetric stretching [43], and the latest additional peak observed at 559 cm^−1^ matched bending vibrations [44]. The broad nature of the peak at 559 cm^−1^, along with the shoulder observed at 1212 cm^−1^, confirms that the biosilica sample is amorphous [45]. The FTIR-ATR spectrum lacks weak bands at 3450 cm^−1^ and 1646 cm^−1^, corresponding to the O-H stretching and angular vibrations of water molecules, respectively. This indicates that the water content in the biosilica sample is low [46].

The obtained average diameter distribution function for the particle sizes of biosilica is presented in Figure 2. The average hydrodynamic diameter was approximately 691 nm for biosilica.

Three types of MWCNTs, with different physical properties, were used in this study. They reinforced the cement mortar using effective dispersion methods. Exposure to MWCNTs improved their mechanical properties and water absorption. The compressive and flexural strengths of the cement mortar can be improved by MWCNTs.

Nanotubes, particularly carbon nanotubes (CNTs), are nanoscale cylindrical structures made of carbon atoms. They possess exceptional mechanical properties, including high tensile strength and stiffness.

When dispersed within the cement, nanotubes can act as nano reinforcements, providing additional strength and stiffness to the material. They can effectively bridge microcracks within the cement matrix, preventing crack propagation and enhancing the material’s fracture toughness.

Nanotubes can also improve the interfacial bonding between cement particles and the surrounding matrix. This enhanced bonding helps to transfer stress more efficiently between the different phases of the material, leading to improved mechanical performance. Additionally, nanotubes can influence the nucleation and growth of hydration products during cement hydration. By serving as nucleation sites for hydration products such as C-S-H gel, nanotubes can help create denser and more uniform microstructures, leading to enhanced mechanical properties [44].

MWCNTs with three different outer diameters and different specific surface areas (SSA) (Chengdu Organic Chemicals Co., Ltd., Chengdu, China) were used in this work. The scanning electron microscope (SEM) images are shown in Figure 3, and the physical properties are presented in Table 4.

### 2.2. Mixing and Sample Preparation

According to the research findings [25], incorporating 10% biosilica into cement mortar mixtures led to a 47.3% increase in compressive strength compared to the control sample. The compositions developed utilized 10% biosilica as an additive in the cement mass, with a water–cement ratio of 0.47 and a cement–sand ratio of 1:2.5 for mixtures. The preparation of the control specimen involved mixing cement and sand for 2 min, followed by adding water and continued mixing for another 3 min, resulting in a total mixing of another 5 min. The mixing was carried out using a mortar mixer, specifically the E095 model produced by MATEST (Treviolo, Italy).

When 10% biosilica was added to the cement mortar composition, the mixing process was conducted in the following order: cement and sand were initially mixed in a mortar mixer for 2 min, then, water and biosilica were added and mixed using a magnetic stirrer (Magnetic stirrer MM-5, Uzhgorod, Ukraine) for 10 min. The water–biosilica solution was gradually added to the dry cement–sand mixture and all components were mixed for 3 min. The total mixing duration for the remaining components was 15 min. Three types of MWCNTs—TNM1, TNM2, and TNM3—were used in the developed compositions, each prepared with a concentration of 0.05, 0.10, and 0.15%. Six prism-shaped test specimens (40 mm × 40 mm × 160 mm) were prepared for the 7- and 28-day tests. Disperbyk, a dispersant, was used to achieve the homogeneous mixture dispersion of the MWCNTs in the mixture. The mortar preparation involved combining cement and sand in a mortar for two minutes. Then, water and MWCNTs were ultrasonicated for thirty minutes using an Ultrasonic machine (UP400St, 400 W, 24 kHz Hielscher digital sonicator, Teltow, Germany). The resulting mixture was transferred to a magnetic stirrer and mixed for an additional three minutes with the addition of biosilica. The water–MWCNTs–biosilica mixture was subsequently added to the dry mixture of cement and sand, which all were mixed for an additional three minutes. The final cement mortar was then molded into standard metal molds with dimensions of 40 mm × 40 mm × 160 mm.

Samples were prepared for testing with concentrations of MWCNTs ranging from 0.05% to 0.15%, adding 10% biosilica and the same concentrations of nanoparticles without biosilica. After 24 h, the specimens were demolded and placed in a storage chamber with a temperature of (20 ± 2) °C and a humidity of (98 ± 2)%. On the 28th day, the specimens were removed from the water and subjected to a compressive strength test. Six cubes with dimensions of 40 mm × 40 mm were tested. The average value of the compressive strengths of the six test specimens was calculated under [37], point 10.2, as the final result.

In Figure 4, the process of preparing the test specimens, raw materials, preparation of the mortar mixture, curing for 7 and 28 days under (20 ± 2) °C and (98 ± 2)% humidity, and determination of compressive strength are presented.

### 2.3. Compressive Strength

The compressive strength method has two primary applications, namely quality control and acceptance testing. This section describes a method of quality testing in which cement is assessed to determine whether it meets the compressive strength requirements. The samples are evaluated through compression testing on their side surfaces that measure 40 mm × 40 mm. Throughout the duration of the load application, the load is progressively increased at a rate of (2400 ± 200) N/s until failure takes place. The compressive strength Rc is calculated in N/mm^2^ using the formula:


(1)
$$Rc=\frac{F_{c}}{1600}$$


where—Rc is the compressive strength, N/mm^2^;Fc—maximum load, N;1600 = (40 mm × 40 mm)—area of slabs, mm^2^.

During the testing process, three random samples were selected from each batch. These samples underwent a strict compressive strength measurement procedure, where an innovative 2000 kN automatic concrete compression machine (Servo-Plus Progress, MATEST, Treviolo, Italy) was used to measure the strength precisely according to EN 196-1 [47] guidelines. The specimens tested for compressive strength had uniform dimensions of 40 × 40 mm, ensuring standardization of the evaluation procedure. The compressive tests were conducted at two critical time points, i.e., 7 and 28 days. The automated compression machine (C089, MATEST) with a specified loading rate of 2.4 kN/s made this evaluation feasible [37,48,49].

### 2.4. Water Absorbtion

The water absorption process for cement involves water penetration into the porous microstructure of the cement matrix. When cement comes into contact with water, the following steps occur: Water molecules begin to infiltrate the surface of the cement particles. This process occurs due to the presence of surface-active sites and capillary action within the porous microstructure of the cement. Capillary forces drive water movement through the interconnected capillaries and voids within the cement matrix. These capillaries act as pathways for water movement, allowing it to penetrate deeper into the material. The water absorption process for cement is a dynamic interaction between water and the porous microstructure of the cement matrix, influenced by factors such as pore size distribution, cement composition, and environmental conditions [50,51].

Water Absorption is determined by the equation:


(2)
$$\mathrm{Water}\;\mathrm{Absorption}\;(\%)=\frac{W_{w}-W_{d}}{W_{d}\times 100}$$


where—W_w_ = Weight of the wet cement sample after immersion, g,W_d_ = Weight of the dry cement sample before immersion, g.

All test specimens composed of components N0–19, in accordance with the guidelines stipulated in GOST 12730.3-2020 [52], were evaluated for their water absorption by mass after 28 days.

### 2.5. Electrical Properties

The four-probe approach using direct current (DC) is widely used to assess the electrical properties of a self-sensing cement matrix. Bulk electrical resistivity measurements of the cement mortar composites were performed utilizing a measuring platform and the four-probe method [35,53]. The measurements were performed for every batch with three samples at 1-day, 7-day, 14-day, and 28-day intervals using a digital multimeter (DMM6500, 6 ½-digit, Keithley Instruments Inc., Solon, OH, USA). The mean values are reported in the findings.

Through cyclic compression testing, the piezoresistive behavior of cement mortars based on MWCNTs was investigated. Electrical measurements were performed simultaneously with the experiments. As seen in Figure 5, a load frame with a capacity of 2000 kN was chosen for the experiments (Italian Matest, C089, 2000 kN automatic Servo-Plus Progress Concrete compression machine, Treviolo, Italy).

MWCNT-based cement mortar samples were subjected to four loading and unloading cycles at each amplitude, or 5, 10, and 15 kN load amplitudes, with loading rates of 0.04, 0.08, and 0.12 kN/s. A DC power source was employed as a power source on an electrical measurement platform to assess the electrical resistivity of the specimens. To solve the voltage differential, one digital multimeter was attached to the specimens’ inner probes; a second multimeter, which measured current intensity, was connected to the outer probes via a power supply unit.

The resistance MWCNT-based cement mortar was computed using the following equation [35,52,53]:
R = V/I(3)
where R is the electrical resistance of the MWCNT-based cement mortar, V is the voltage, and I is the current provided. The resistivity of the composite samples was calculated as follows:
ρ = (R⋅S)/L(4)
where ρ is the resistivity of the composite specimens, S is the surface of the cross-section of the samples, and L is the interspace of the inner probes. The resistivity measurements were carried out on days 1, 7, 14, and 28 for all batches containing three samples, and the mean values are indicated in the results.

To assess the piezoresistivity of MWCNT-based cement mortar samples, the fractional change in resistivity (FCR) was computed as:FCR = (ρ_t_ − ρ_o_)/ρ_o_(5)
where ρ ρ_t_ is the resistivity at t time through the cyclic compressive test [52,53].

## 3. Results and Discussion

The data in Table 5 display the values of density, water absorption, and compressive strength for the test samples after 7 and 28 days of curing.

The analysis performed on the basis of the data obtained is presented in Section 3.1.

### 3.1. Mechanical and Physical Properties

Analyzing the data presented in Figure 6, it can be concluded that the compressive strengths after 7 days with MWCNTs1 increased in the cases of 0.05% and 0.1% (composites N2 and N3) compared to the compressive strengths of the reference sample (composition N0), which were 1.3% and 6.4%, respectively. Moreover, in the case of the 0.15% concentration (N4 composition), it decreased by 10.3%. After 28 days of curing, in the cases of 0.05% and 0.1% concentrations, they increased by 6.8 and 21.7%, and in the case of 0.15%, they decreased by 8.9%. Upon evaluating the information depicted in Figure 6, it can be inferred that the compressive strengths after 7 days with MWCNTs1 exhibited an increase for the 0.05 and 0.1% concentrations (composites N2 and N3) compared to the compressive strength of the reference sample (composition N0), with respective increments of 1.3% and 6.4%. Conversely, for the 0.15% concentration (N4 composition), there was a decrease of 10.3% in compressive strength. After 28 days of curing, the compressive strengths for the 0.05 and 0.1% concentrations (composites N2 and N3) showed increases of 6.8% and 21.7%, respectively, while the 0.15% concentration (N4 composition) experienced a decrease of 8.9%.

The compressive strengths of samples prepared with MWCNTs2 at concentrations of 0.05, 0.1, and 0.15% increased by 54.9, 74.3, and 53.2%, respectively, in the 7-day test. Compared to the reference sample, the compressive strengths at concentrations of 0.05 and 0.1% increased by 0.8% and 3.5% in the 28-day test, while at 0.15%, they decreased by 8.1%. In contrast, the compressive strengths of samples prepared with MWCNTs3 at concentrations of 0.05 and 0.15% decreased by 0.6% and 6.5%, respectively, in the 7-day test, and increased by 10.8% at 0.1% concentrations. The results of the 28-day test indicated that the compressive strengths of the samples made with 0.05% and 0.1% (N14 and N15 compositions) had increased by 0.6% and 4.2%, respectively, but had decreased by 11.9% when the concentration was 0.15% (Figure 6).

The compressive strengths of the test samples with 0.05 and 0.1% MWCNTs1 and 10% biosilica (compositions N5 and N6) after 7 days outperformed the values of the samples with N1 composition by 2.6% and 6.7%, respectively. However, the compressive strength of the samples with 0.15% MWCNTs1 and 10% biosilica (composition N7) decreased by 19.6% compared to the samples with N1 composition. In 28 days, the values for components N5 and N6 increased by 2.5% and 8.5%, respectively, while the value for component N7 decreased by 17%. After preparing the samples with MWCNTs2 and adding 10% biosilica, the compressive strength values after 7 and 28 days of curing were as follows: for compositions N11–N13, the strengths of 0.05, 0.1, and 0.15% MWCNTs2 increased by 19.1%, 29.2%, and 14.4%, respectively, at 7 days; at 28 days, the strengths at 0.05 and 0.1% MWCNTs2 concentrations (compositions N11 and N12) increased by 5.6% and 12.6%, respectively, while the strength at the 0.15% concentration (composition N13) decreased by 10.6%. For compositions N17 and N19 with 0.05 and 0.15% MWCNTs3 concentrations and 10% biosilica, the compressive strengths decreased by 0.3% and 24.7%, respectively, while for composition N18 with the 0.1% MWCNTs3 concentration, the strength increased by 2.1% after 7 days. After 28 days, the compressive strength of the specimens made with compositions N17 and N18 increased by 1.8% and 6.3%, respectively, while the strength of the specimens made with composition N19 decreased by 21.1%. All test samples prepared with MWCNTs and 10% biosilica (compositions N2–N19) were compared with the compressive strength value of the reference sample prepared with the N1 composition (Figure 7).

Upon evaluating the outcomes of the compressive strength assessments, it is evident that in each instance, the strength values were enhanced with the incorporation of 0.05 and 0.1% MWCNTs, while in the case of 0.15%, they experienced a decrease. Similarly, identical results were obtained when 10% biosilica was combined with MWCNTs. Furthermore, upon examining the water absorption test results, it was revealed that the indicators diminished for nearly all concentrations of MWCNTs.

Here, we show our findings on water absorption for samples with MWCNTs and those with MWCNTs and 10% biosilica.

At 28 days of age, the water absorption of test samples made with cement mortar and MWCNTs1, MWCNTs2, and MWCNTs3 concentrations of 0.05, 0.1, and 0.15% (by cement mass) was determined and compared with the water absorption result of the reference (N0) sample [52]. The results of the comparison showed that in the case of MWCNTs1 at 0.05, 0.1, and 0.15% concentrations (N2–4 compositions), the water absorption of the test samples with 0.05% MWCNTs1 increased by 1.1% compared to the water absorption of the reference sample; in the case of 0.1%, it decreased by 3.2%, and in the case of 0.15% concentration, it remained the same. Similarly, in the case of MWCNTs2 at 6.8, 11.8, and 19% (compositions N8–10), the water absorption of the test samples decreased by 1.1, 5.6, and 3.3%, respectively. Additionally, in the case of MWCNTs3 at 0.05, 0.1, and 0.15% concentrations (compositions N14–16), the water absorption of the test samples decreased by 0.6, 1.9, and 1.4%, respectively (Figure 8).

When MWCNTs were added to the cement mortar, 10% biosilica was also added, and test samples were prepared. The water absorption values of these samples were compared to those of the test samples with N1 composition. The results are as follows: for MWCNTs1 at a concentration of 0.05% (N5 composition), water absorption increased by 1.6%. However, at concentrations of 0.1% and 0.15% (components N6 and N7), water absorption values decreased by 3.5% and 4.7%, respectively. For MWCNTs2, water absorption values decreased by 7.2%, 17.1%, and 21.9% at concentrations of 0.05%, 0.1%, and 0.15%, respectively (components N11–13). Additionally, when test samples were prepared with 10% biosilica and MWCNTs3 concentrations of 0.05%, 0.1%, and 0.15% (compositions N17–19), the results showed a decrease of 7.2%, 12.6%, and 18.7%, respectively (Figure 8).

MWCNTs improve the durability of concrete by reducing water absorption, preventing the ingress of harmful substances such as chloride ions and sulphates, and enhancing resistance to chemical attack and corrosion. The hydrophobic nature of MWCNTs can contribute to reducing the water absorption and permeability of cementitious materials. MWCNTs assist in reducing early-age cracking in concrete by enhancing its tensile strength and ductility. These nanotubes can speed up the cement hydration process by serving as nucleation sites for hydration products and aiding in the formation of calcium silicate hydrate (C-S-H) gel. This promotes the creation of denser and more uniform cementitious structures, leading to enhanced mechanical properties and durability [54].

The density of mortar is the measurement of the combined mass of cement, sand, and water, uniformly mixed and agitated. The density, which is directly proportional to the components of the mortar, directly influences the strength of the mortar. As the density increases, water absorption decreases and mortar strength increases. Eliminating easily washed-out calcium hydrosilicate from the composition enhances the compaction of the solution matrix, thereby increasing the density of the cement mortar and the durability of the stone. The samples prepared with MWCNTs and biosilica exhibited a higher density than the reference sample. The cause of this issue lies in the fact that these particles are composed of fine particles, unlike cement particles. This results in a denser structure because the smaller particles fit into the spaces left by the larger particles. A reduction in pore size and an increase in cement stone density are achieved. The additional gel of calcium silicate hydrate (CSH) produced by the reaction enhances the density and connectivity of the cementitious matrix. This densification further restricts the movement of water through the concrete. Additionally, the formation of the CSH gel and cement densification decrease the connected pore structure. This reduction in connecting routes restricts movement, which lowers the concrete’s permeability [55,56].

### 3.2. Bulk Electrical Resistivity

To assess the impact of MWCNT concentrations (0.05, 0.10, and 0.15 weight percent) on the electrical resistivity of MWCNT-reinforced cement mortar, Figure 9 shows the relationship plots of resistivity with different MWCNT doses at 1-day, 7-day, 14-day, and 28-day curing ages.

The electrical resistivity of MWCNT cement composites increases with aging, as illustrated in Figure 8. This is because the cement hydration process reduces the amount of pore water in the sample’s microstructure. Given that water has high electrical conductivity [52], the electrical resistance rises as the amount of pore water decreases. After seven days, the cement hydration reaction slows down, leading to a slower rate of increase in resistivity.

The various types of MWCNTs and the sizes of nanotubes can impact the cement hydration process [57,58]. The graphs of the resistivity-MWCNT ratio at 1- and 7-days post-curing indicate that resistivity rises as the MWCNT dosage increases. Electrical resistance usually decreases as the concentration of conductive particles increases. The curve’s structure changed after 14 days. The resistivity reaches its lowest point at a concentration of 0.05% and then starts to increase. This is due to the potential for nanoparticle aggregation, which occurs as the sample particle dose increases, as well as the slow cement hydration process.

### 3.3. Cyclic Compressive Tests Results

In this part, the piezoresistive characteristics of cement mortar samples reinforced with MWCNT at different concentrations (0.05, 0.10, and 0.15 wt.%) are investigated by conducting cyclic compressive tests. To obtain the electrical measurements, DC was utilized, and the results are presented. The FCR of the cement mortar samples reinforced with 0.05, 0.10, and 0.15 percent MWCNT, respectively, is illustrated in Figure 10.

According to the graphs, it is evident that all specimens exhibit piezoresistive properties, and the measurements reveal a trend in FCR, indicating that the initial resistivity of the specimens increases continuously with cyclic loading. However, the 0.15% concentration batch (third batch), especially the third sample, exhibits a different curve structure. In contrast, as the load amplitude increases, FCR decreases. The relationship between the continuous increase in initial resistivity and the reduction in compressive strength of the samples can be attributed to the emergence of microcracks within the samples, which simultaneously increase electrical resistivity. This is evident in the FCR results (Figure 10). Conversely, the third sample from the third batch exhibits a continuous decrease in initial resistivity. This decrease can be linked to the densification of the particles to each other without any formation of damages or cracks within the specimen.

For the first and second batches, all samples have the same curve structure and better piezoresistive response with a deviation in FCR.

## 4. Conclusions

This study investigated the combined influence of various diameters and concentrations of multi-walled carbon nanotubes on the water absorption, compressive strength, and electrical properties of cement mortar. A total of 19 compositions of cement mortars, both with and without biosilica, were utilized in the experiments. Based on the studies conducted and the outcomes obtained, the following conclusions can be drawn:

The 7-day compressive strength of samples prepared without biosilica was as follows: MWCNT1 was 34.7–39.2 MPa, MWCNT2 was 59.3–67.5 MPa, and MWCNT3 was 36.2–42.9 MPa, while the compressive strength of the reference sample was 38.7 MPa. When 10% biosilica was used in the cement mortar formulations, the 7-day compressive strength of the test specimens was as follows: MWCNT1 was 42.6–56.6 MPa, MWCNT2 was 60.7–68.5 MPa, and MWCNT3 was 39.9–54.2 MPa, while the strength of the reference sample was 53.0 MPa.In the results of the 28-day compressive strength test, the following data were obtained:-Prototypes prepared from MWCNT1 exhibited a strength of 62.2–83.2 MPa.-Prototypes prepared from MWCNT2 exhibited a strength of 62.8–68.9 MPa.-Prototypes prepared from MWCNT3 exhibited a strength of 60.1–71.2 MPa.The compressive strength of the reference sample was 68.3 MPa. These results were obtained for compositions without biosilica. For compositions with biosilica, the following surface strengths were observed—MWCNT1: 68.0–89.0 MPa; MWCNT2: 73.3–92.4 MPa; MWCNT3: 64.6–87.1 MPa The strength of the reference sample was 82.0 MPa. The highest compressive strength was observed in the case of composites processed with a 0.1% concentration of MWCNTs2, while the strengths for 0.05% and 0.15% concentrations were relatively low. At 28 days, the strength of samples without biosilica increased by 29.2%, and in the case of biosilica, it increased by 12.6%.At a concentration of 0.05%, the electrical resistivity reached its lowest value, possibly due to the agglomeration of nanoparticles. The curves show that all specimens exhibited piezoresistive properties, and the measurements indicated a continuous increase in initial resistivity with cyclic loading, except for the batch of specimens with a concentration of 0.15% (3rd batch). Specifically, the third sample in this batch showed a change in the curve structure. For the first and second batches, all samples had almost the same curve structure and better piezoresistive response with a deviation in FCR. At a concentration of 0.05%, the electrical resistivity reached its lowest value, possibly due to the agglomeration of nanoparticles. The graphs demonstrate that all samples showed piezoresistive properties, and the measurements indicated a consistent increase in initial resistivity with cyclic loading, except for the specimens with a concentration of 0.15% (third batch). Particularly, the third sample in this batch exhibited a change in the curve structure. For the first and second batches, all samples had nearly identical curve structures and displayed better piezoresistive responses with a variation in FCR.

## Figures and Tables

**Figure 1 nanomaterials-14-01271-f001:**
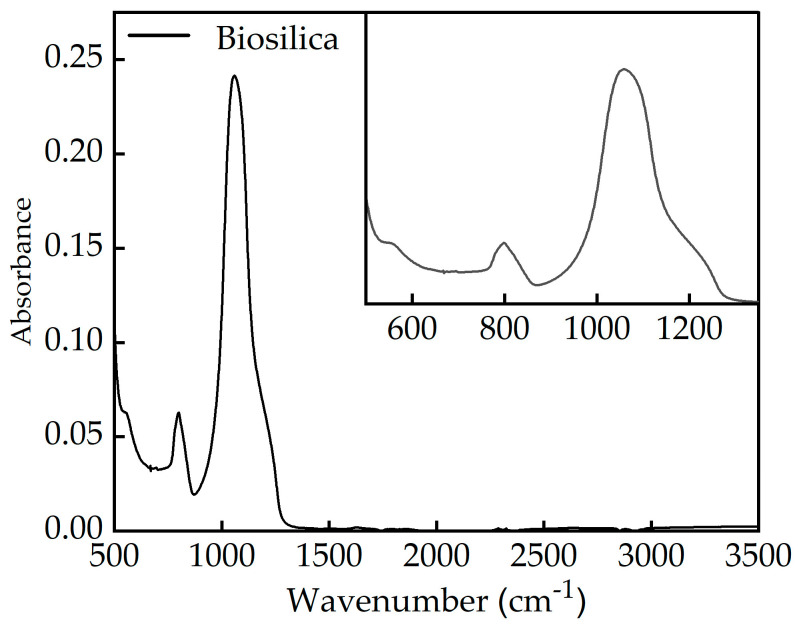
FTIR-ATR spectrum of a biosilica sample.

**Figure 2 nanomaterials-14-01271-f002:**
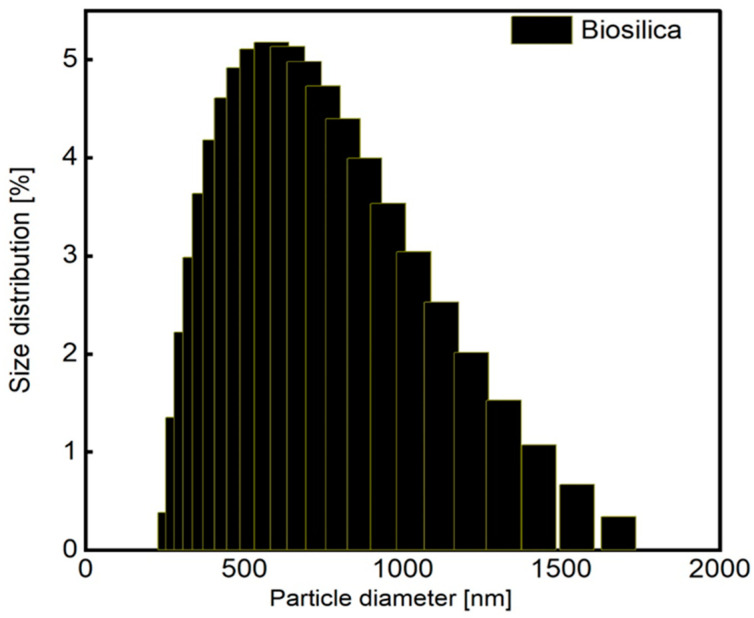
Particle size distribution of biosilica.

**Figure 3 nanomaterials-14-01271-f003:**
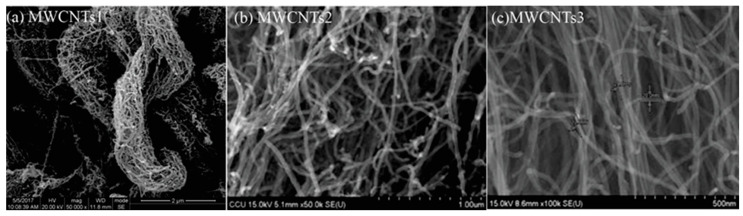
SEM images of MWCNTs: (**a**) MWCNTs1, (**b**) MWCNTs2, and (**c**) MWCNTs3.

**Figure 4 nanomaterials-14-01271-f004:**
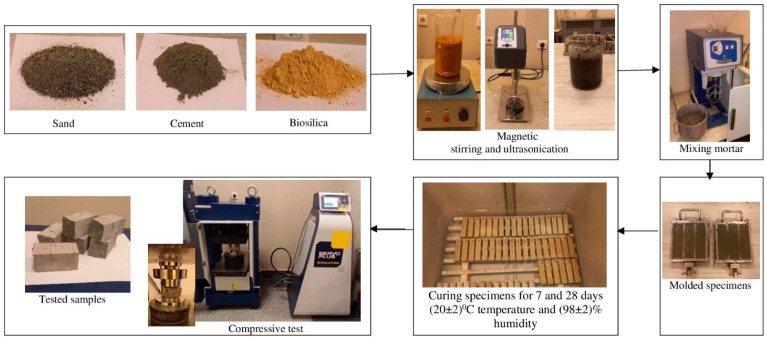
Sample preparation process sequence and testing procedure.

**Figure 5 nanomaterials-14-01271-f005:**
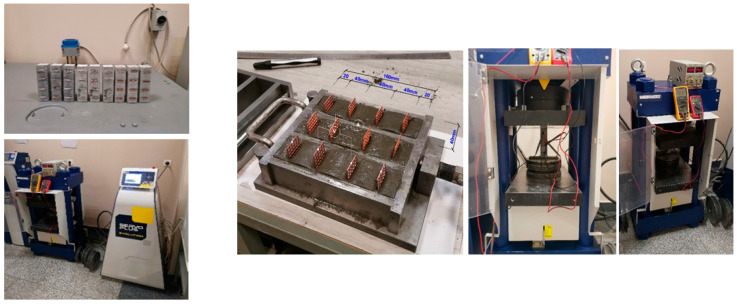
Prismatic samples with copper mesh sheets and Cyclic compression setup.

**Figure 6 nanomaterials-14-01271-f006:**
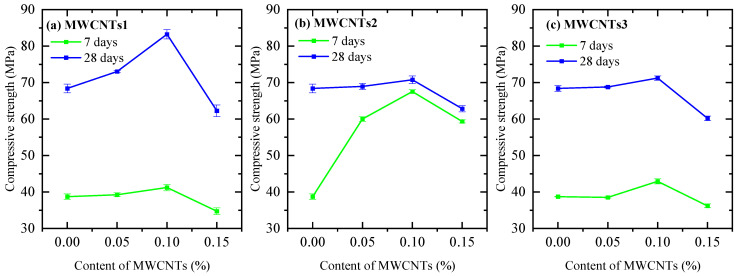
Seven- and twenty-eight-day compressive strength data of the test samples prepared with MWCNTs1 (**a**), MWCNTs2 (**b**), and MWCNTs3 (**c**).

**Figure 7 nanomaterials-14-01271-f007:**
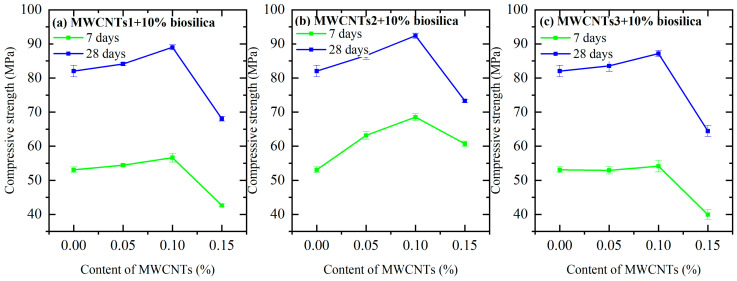
Compressive strength data after 7 and 28 days prepared with MWCNTs1 (**a**), MWCNTs2 (**b**), MWCNTs3 (**c**), and 10% biosilica.

**Figure 8 nanomaterials-14-01271-f008:**
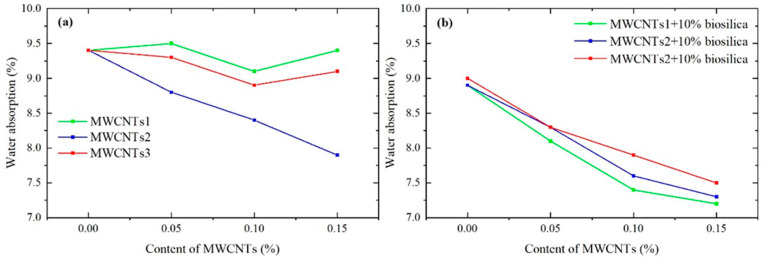
Water absorption results of (**a**) MWCNTs(1–3) and (**b**) MWCNTs(1–3) + 10% biosilica samples at 28 days.

**Figure 9 nanomaterials-14-01271-f009:**
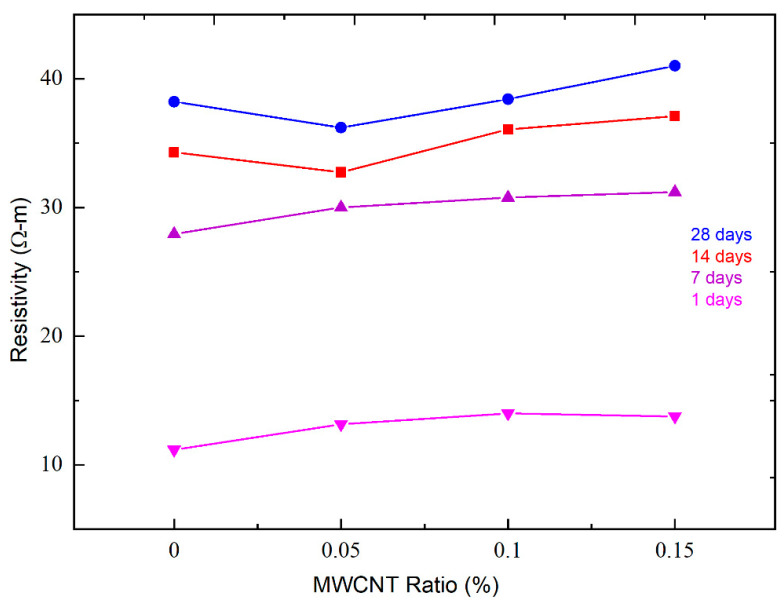
Bulk resistivity of cement mortars with different MWCNT concentration levels at 1, 7, 14, and 28-day curing ages.

**Figure 10 nanomaterials-14-01271-f010:**
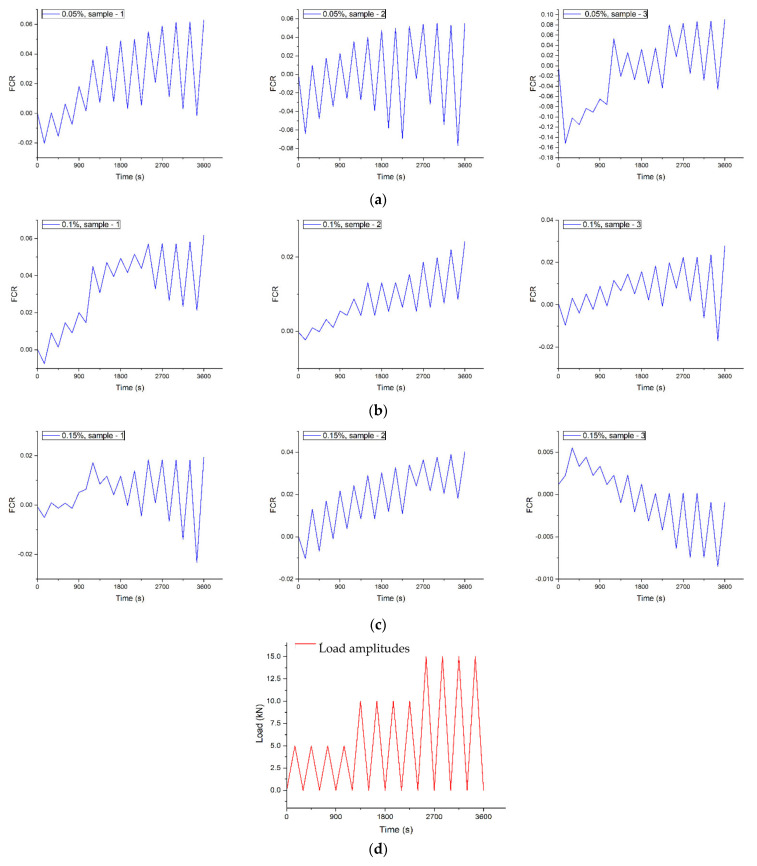
The FCR under compressive load for 28-day samples with different MWCNT concentrations. (**a**) 0.05%, (**b**) 0.10%, and (**c**) 0.15 wt.%. Applied cyclic compressive 5, 10, and 15 kN load amplitudes with loading rates of 0.04, 0.08s, and 0.12 kN/s (**d**).

**Table 1 nanomaterials-14-01271-t001:** Physical and mechanical properties and chemical composition of cement.

Characteristics				Unit			Results Obtained	
**Specific gravity**				g/cm^3^			3.1	
**Blain’s fineness**				cm^2^/g			3250	
**Standard consistency**				%			30.2	
**Setting time**		Initial		min			45	
		Final					285	
**Compressive strength**		3 days		MPa			20	
		7 days					38	
		28 days					52	
**Chemical composition of cement (wt.%)**								
**Al_2_O_3_**	**SiO_2_**	**Fe_2_O_3_**	**CaO**	**MgO**	**SO_3_**	**Free** **CaO**	**Insol. resid.**	**Loss on ignition**
4.16	28.1	3.25	53.4	3.8	1.9	1.09	2.5	1.8

**Table 2 nanomaterials-14-01271-t002:** Average data on the physical properties of sand.

Sieve Residues, %					Size Modulus, Mк	Specific Gravity, g/cm^3^	Bulk Density in Compact State, kg/m^3^	Bulk Density in Loose State, kg/m^3^
2.5	1.25	0.63	0.315	0.16				
17.34	32.16	53.46	75.52	95.68	2.7	2.5	1800	1670

**Table 3 nanomaterials-14-01271-t003:** Chemical composition (wt. %) and physical properties of biosilica.

Chemical Composition (wt.%)				
SiO_2_	Al_2_O_3_	Fe_2_O_3_	K_2_O	MgO
88.92	6.1	2.8	1.34	0.84
**Physical properties**				
**Residue on sieve 45 µm, %**		**Bulk density, kg/m^3^**		
5.0		280		

**Table 4 nanomaterials-14-01271-t004:** Physical properties of MWCNTs.

MWCNT	Outer Diameter	Length	SSA	Purity
MWCNTs1	4–6 nm	10–20 µm	>380 m^2^/g	>98%
MWCNTs2	5–15 nm	10–30 µm	>220 m^2^/g	>98%
MWCNTs3	20–30 nm	10–30 µm	>110 m^2^/g	>98%

**Table 5 nanomaterials-14-01271-t005:** The composition, physical, and mechanical properties of the cement mortar.

N	Cement, g	Sand, g	W/C	Biosilica, g	MWCNTs, %	Disperbyk, g	Density, g/cm^3^	Water Absorption, %	Compressive Strength, MPa	
									7 Days	28 Days
**0**	880	2200	0.47	-	-	-	2.14	9.4	38.74	68.39
**1**	880	2200	0.47	88	-	-	2.19	8.9	53.06	82.04
**MWCNT1**										
**2**	880	2200	0.47	-	0.05	0.44	2.16	10.5	39.26	73.05
**3**	880	2200	0.47	-	0.1	0.88	2.20	10.8	41.22	83.24
**4**	880	2200	0.47	-	0.15	1.32	2.20	10.6	34.73	62.27
**5**	880	2200	0.47	88	0.05	0.44	2.20	9.05	54.44	84.13
**6**	880	2200	0.47	88	0.1	0.88	2.20	8.6	56.64	89.01
**7**	880	2200	0.47	88	0.15	1.32	2.19	8.5	42.62	68.06
**MWCNT2**										
**8**	880	2200	0.47	-	0.05	0.44	2.18	8.8	60.01	68.96
**9**	880	2200	0.47	-	0.1	0.88	2.20	8.4	67.54	70.77
**10**	880	2200	0.47	-	0.15	1.32	2.21	7.9	59.35	62.84
**11**	880	2200	0.47	88	0.05	0.44	2.21	8.3	63.18	86.62
**12**	880	2200	0.47	88	0.1	0.88	2.21	7.6	68.54	92.41
**13**	880	2200	0.47	88	0.15	1.32	2.20	7.3	60.70	73.30
**MWCNT3**										
**14**	880	2200	0.47	-	0.05	0.44	2.15	9.3	38.51	68.80
**15**	880	2200	0.47	-	0.1	0.88	2.18	8.9	42.92	71.24
**16**	880	2200	0.47	-	0.15	1.32	2.18	9.1	36.21	60.19
**17**	880	2200	0.47	88	0.05	0.44	2.20	8.3	52.91	83.54
**18**	880	2200	0.47	88	0.1	0.88	2.21	7.9	54.15	87.19
**19**	880	2200	0.47	88	0.15	1.32	2.20	7.5	39.94	64.66

## Data Availability

The data presented in this study are available upon request from the corresponding author.

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
