# Peer review of "Simultaneous Effect of Diameter and Concentration of Multi-Walled Carbon Nanotubes on Mechanical and Electrical Properties of Cement Mortars: With and without Biosilica"

_nanomaterials, 2024, doi:10.3390/nano14151271_

Round 1

Reviewer 1 Report

Comments and Suggestions for Authors

This study intends to investigate the effects of multiwalled carbon nanotubes (0.05, 0.10, and 0.15%) with varying outer diameters and specific surface areas on the mechanical and electrical properties of cement mortar both with and without biosilica (10% by weight of cement). It is worth noting that the obtained results were intended to provide insightful insights into optimizing biosilica and multi-walled carbon nanotube incorporation in cement mortar. To contribute to the development of building materials, this research delves into the intriguing application of biosilica and multi-walled carbon nanotubes in cement mortar, highlighting the important influence of mixing techniques on the piezoresistive properties. The manuscript can be further improved by a major revision by addressing the following comments:

1.      It is required to check the entire manuscript to improve the English language since several distinctly linguistic errors can be identified in the text, for instance:

- line 183: “…indicating a combination of asymmetric stretching …” which may be changed to “… indicates …”,

- line 228: “…used in this work, which the scanning electron microscope…” which may be changed to “…used in this work, and the scanning electron microscope…”,

- lines 319, 336 and 353:  square brackets should be corrected,

- line 535: Figure 56,

- line 594: “…structure and a better piezoresistive…” which may be changed to “…structure and better piezoresistive…”,

- Figure 4: no information in the text of the manuscript.

2.      The References section requires modification, and please check "Referencing style and Abbreviated Journal Name" in all references following the “Nanomaterials” Instructions for Authors.

3.      Information related to the use of ultrasonication (lines 18 and 256) should be detailed, for instance:

- Ultrasonic bath or sonicator was used?

- Power and frequency of the ultrasonication?

- Because ultrasonic treatment can significantly affect the hydration process and enhance the mechanical and microstructural properties of cementitious composites (https://www.sciencedirect.com/science/article/pii/S0264127518307457), control tests without ultrasonication on cement mortar preparation are recommended.

4.      Thermal, macroscopic, and microstructural characteristics are crucial indicators for assessing how well multi-walled carbon nanotubes and biosilica have been incorporated into cement mortar. Using readily available techniques (e.g., TGA/DTA/DSC, XRD, MIP, Isothermal Calorimetry) that were absent from this research paper, it is possible to make up for the lack of information in the current manuscript regarding the properties of prepared cementitious composites. This could serve as a useful reference for the comprehensive evaluation of new cement-based materials.

Comments on the Quality of English Language

Moderate editing of English language required

Author Response

Response to Reviewer 1

Dear Reviewer

Thank you very much for your careful and constructive comments. Your comments are very helpful for improving our paper. According to your recommendations, we revised the manuscript and made corrections which we hope meet with approval, and the modified parts were all marked in red. We hope you are satisfied with the revised version.

Point 1. It is required to check the entire manuscript to improve the English language since several distinctly linguistic errors can be identified in the text, for instance:

- line 183: “…indicating a combination of asymmetric stretching …” which may be changed to “… indicates …”,

- line 228: “…used in this work, which the scanning electron microscope…” which may be changed to “…used in this work, and the scanning electron microscope…”,

- lines 319, 336 and 353:  square brackets should be corrected,

- line 535: Figure 56,

- line 594: “…structure and a better piezoresistive…” which may be changed to “…structure and better piezoresistive…”,

- Figure 4: no information in the text of the manuscript.

Response 1. Based on the comments, all changes are implemented.

Point 2. The References section requires modification, and please check "Referencing style and Abbreviated Journal Name" in all references following the “Nanomaterials” Instructions for Authors.

Response 2. According to “Nanomaterials” Instructions for Authors, all changes are implemented.

   Point 3.   Information related to the use of ultrasonication (lines 18 and 256) should be detailed, for instance:

Response 3.

- Ultrasonic bath or sonicator was used?

Was used Ultrasonic machine.

- Power and frequency of the ultrasonication?

The required information is added to the manuscript.

- Because ultrasonic treatment can significantly affect the hydration process and enhance the mechanical and microstructural properties of cementitious composites (https://www.sciencedirect.com/science/article/pii/S0264127518307457), control tests without ultrasonication on cement mortar preparation are recommended.

Thanks for the recommendation. We will definitely take it into account in our future research.

   Point 4. Thermal, macroscopic, and microstructural characteristics are crucial indicators for assessing how well multi-walled carbon nanotubes and biosilica have been incorporated into cement mortar. Using readily available techniques (e.g., TGA/DTA/DSC, XRD, MIP, Isothermal Calorimetry) that were absent from this research paper, it is possible to make up for the lack of information in the current manuscript regarding the properties of prepared cementitious composites. This could serve as a useful reference for the comprehensive evaluation of new cement-based materials.

Response 4. Due to lack of time, we cannot carry out the mentioned research, but we will do it in the future research.

Faithfully yours,

Dr. Manuk Barseghyan (on behalf of all authors)

National University of Architecture and Construction of Armenia, Armenia

Reviewer 2 Report

Comments and Suggestions for Authors

This manuscript reports a study on the effects of multi-walled carbon nanotubes on mechanical and physical properties of cement mortars with and without biosilica.  The paper was poorly prepared and not proofread by the authors as stated. Five individuals were indicated as contributing to the writing and review of the paper--obvious errors indicate this was a misrepresentation of the facts.  

Much work on functionalization of MWCNTs with carboxylic acid, hydroxyl and other groups exists, but rather than including a relevant paragraph in the Introduction on this topic for nanotubes, one on functionalization of graphene oxide is arcanely provided. While the results are interesting, the approach is mainly a description of what was done and what was observed with little effort to look deeper into the why or into comparisons to work by others.   For example, all their sets of experiments on compressive strength show an increase followed by a decrease as the percentage of MWCNTs is raised from zero.   Many papers in this topic area attribute this to agglomeration of the nanotubes, but this explanation nor any other is given to rationalize their results.  Contrast your result with that of others.  Findings on mechanical properties for nanotubes of different sizes and aspect ratios should be compared with the literature (See Cui et al., 2017; Manzur et al. 2014; and others).

The description of results for Figure 8 in the paragraph under the figure doesn't accurately represent what's shown.   For example, water absorption for MWCNs1 without biosilica is said to decrease while the slope is clearly positive.  Values in the text aren't consistent with the percentages shown in the figure.

Lines 22-25: For what concentration of MWCNTs?

Line 60: .... with an optimal volume fraction of ....    As your results and those of others show, a higher volume fraction often leads to poorer properties.

Line 121: missing period

Line 129, Table 1:  Uniform capitalization in title.

Line 133, Tabel 2: Uniform capitalization.

Line 155: missing period

Lines 210-212:  This does not belong in the Experimental section.

Line 216-226: This does not belong in the Experimental section.

Line 227: capitalization

Figure 3:  Include procedures for SEM sample preparation and imaging equipment and conditions.

Section 2.2:   The preparation of samples is ambiguous and needs to be rewritten.  It will help to incorporate Table 5 with compositions in this section.

Line 319: brackets?

Line 326: insert space between value and units

Line 336: brackets

Line 353:  delete minus sign; brackets

Table 5:  There were no values for compressive strength or absorption in the Table 5. Create a new Table 6 with water absorption and compressive strength.

Figure 7:  resolution poor

Line 437-438: The informal style is not appropriate.

Comments on the Quality of English Language

English overall is satisfactory.

Author Response

Response to Reviewer 2

Dear Reviewer

Thank you very much for your careful and constructive comments. Your comments are very helpful for improving our paper. According to your recommendations, we revised the manuscript and made corrections which we hope meet with approval, and the modified parts were all marked in green. We hope you are satisfied with the revised version.

Point 1. The description of results for Figure 8 in the paragraph under the figure doesn't accurately represent what's shown.   For example, water absorption for MWCNs1 without biosilica is said to decrease while the slope is clearly positive. Values in the text aren't consistent with the percentages shown in the figure.

Response 1. The following changes are made: The results of the comparison showed that in the case of MWCNTs1 at 0.05, 0.1, and 0.15% concentrations (N2-4 compositions), the water absorption of the test samples with 0.05% MWCNTs1 increased by 1.1% compared to the water absorption of the reference sample, in the case of 0.1%, it decreased by 3.2%, and in the case of 0.15% concentration, it remained the same.

Point 2. Lines 22-25: For what concentration of MWCNTs?

Response 2. The maximum increase in compressive strength was observed in the composites treated with 0.1% MWCNT concentration, and in the case of 0.05 and 0.15% concentrations, the compressive strengths were relatively low.

Point 3. Line 60: .... with an optimal volume fraction of ....    As your results and those of others show, a higher volume fraction often leads to poorer properties.

Response 3. Appropriate changes have been made.

Point 4. Line 121: missing period

Response 4. It is improved.

Point 5. Line 129, Table 1:  Uniform capitalization in title.

               Line 133, Tabel 2: Uniform capitalization.

               Line 227: capitalization

Response 5. All the uniform capitalization in titles are changed.

Point 6. Line 155: missing period.

Response 6. This comment is not understandable.

Point 7. Lines 210-212:  This does not belong in the Experimental section.

Response 7. The mentioned lines are written in section 2.1. Materials.

Point 8. Line 216-226: This does not belong in the Experimental section.

Response 8. The mentioned lines are written in section 2.1. Materials.

Point 9. Figure 3:  Include procedures for SEM sample preparation and imaging equipment and conditions.

Response 9. The MWCNT is made by the Chemical vapor deposition CVD method. As it is mentioned in the manuscript, Chengdu Organic Chemicals Co. was used in this work. Unfortunately, there is no information about the exact name of the equipment of SEM.

Point 10. Section 2.2:  The preparation of samples is ambiguous and needs to be rewritten. It will help to incorporate Table 5 with compositions in this section.

Response 10. The sample preparation process in Section 2.2 is rewritten and the required information is added to Table 5.

Point 11. Line 319: brackets?

                Line 336: brackets

                Line 353:  delete minus sign; brackets

Response 11. Square brackets are corrected.

Point 12. Line 326: insert space between value and units

Response 12. Space between value and units is inserted.

Point 13. Table 5:  There were no values for compressive strength or absorption in the Table 5. Create a new Table 6 with water absorption and compressive strength.

Response 13. Water absorption and compressive strength indicators have been added to Table 5.

Point 14. Figure 7:  resolution poor

Response 14. It is improved.

Point 15. Line 437-438: The informal style is not appropriate.

Response 15. It is improved.   

Point 16. Much work on functionalization of MWCNTs with carboxylic acid, hydroxyl and other groups exists, but rather than including a relevant paragraph in the Introduction on this topic for nanotubes, one on functionalization of graphene oxide is arcanely provided. While the results are interesting, the approach is mainly a description of what was done and what was observed with little effort to look deeper into the why or into comparisons to work by others. For example, all their sets of experiments on compressive strength show an increase followed by a decrease as the percentage of MWCNTs is raised from zero. Many papers in this topic area attribute this to agglomeration of the nanotubes, but this explanation nor any other is given to rationalize their results. Contrast your result with that of others. Findings on mechanical properties for nanotubes of different sizes and aspect ratios should be compared with the literature (See Cui et al., 2017; Manzur et al. 2014; and others).

Response 16. Thank you very much for the helpful suggestion. Due to the short time of the revision of the manuscript, we will take it into account for our next works.

Faithfully yours,

Dr. Manuk Barseghyan (on behalf of all authors)

National University of Architecture and Construction of Armenia, Armenia

Reviewer 3 Report

Comments and Suggestions for Authors

Comments on the Quality of English Language

Author Response

Response to Reviewer 3

Dear Reviewer

Thank you very much for your careful and constructive comments. Your comments are very helpful for improving our paper. According to your recommendations, we revised the manuscript and made corrections which we hope meet with approval, and the modified parts were all marked in yellow. We hope you are satisfied with the revised version.

Point 1. The Introduction needs to be greatly revised. The subject of current study is CNTs, however, graphene oxide ant its applications were introduced in the Introduction. In the revision, literature review of CNTs incorporated cement composites should be presented. In addition, the research gap and significance need to be highlighted.

Response 1. Appropriate changes have been made.

Point 2. P2. Line 57. “The dispersion of nanomaterials uniform distribution of nanomaterials in the cement matrix”. This may not be completely correct. The fresh environment in cementitious composites is a highly alkaline solution, containing mostly calcium cations. The crucial step is to disperse nanomaterials in this calcium saturated alkaline solution. For the GO dispersion in cement paste, the authors can refer to Incorporation of graphene oxide and silica fume into cement paste: A study of dispersion and compressive strength.

Response 2. Appropriate changes have been made.

Point 3. P2, line 76. “changing the crystalline structure of CSH at the nanoscale” This may not be correct.   No solid prove has been found to support this. In addition, “CSH” should be “C-S-H”. Authors should compare previous findings to verify the statement, such as. The early-age cracking sensitivity, shrinkage, hydration process, pore structure and micromechanics of cement-based materials containing alkalis with different metal ions Stress relaxation properties of calcium silicate hydrate: a molecular dynamics study Hydration and fractal analysis on Low-heat Portland cement pastes by thermodynamic- based methods.

Response 3. Appropriate changes have been made.

Point 4. P2, line 81. “Hydroxyl(-OH) from a GO monolayer”. This sentence does not make sense. Functional groups are part of graphene oxide, they can not form a monolayer.

Response 4. Appropriate changes have been made.

Point 5. P2, line 84. “Denser hydration products on the aggregate to the nucleation action of GO”. What is the aggregate? Why hydration products contributed to nucleation effect of GO? The nucleation effect leads to production of more hydration products, not the other way.

Response 5. Appropriate changes have been made.

Point 6. P4, line 147 “capillary reaction products thicken the structure”. What is capillary reaction and capillary reaction products?

Response 6. GO provides additional nucleation sites for the formation of hydration products due to its large surface area and the presence of oxygen-containing functional groups. These nucleation sites promote the growth of more hydration products within the cement matrix.  As more hydration products form, they start to fill the capillary pores within the cement matrix. The capillary pores are the spaces between the hydrated cement particles and aggregates that originally contained water. The formation of hydration products within these pores reduces the overall porosity of the cement matrix. As the hydration products fill the capillary pores, the overall volume of these pores decreases. Reduced capillary porosity means fewer pathways for water and other aggressive agents to penetrate the cement matrix. This contributes to a denser and less permeable cement structure.

Appropriate changes have been made.

Point 7. Line 175-181 are repeated. In addition, the authors are suggested to provide literature references for FT-IR results.

Response 7. We removed repeated lines from 175 to 181.

Point 8. As shown in Table 5, the content was kept un-changed when both biosilica and CNTs were used, compared with samples with CNTs only. When both biosilica and CNTs were used, how was the workability? Normally when more nanomaterials were used, the workability would be lower, which can lead to un-expected air void formation. This would influence the compressive strength. The authors are suggested to explain this.

Response 8. The use of nanomaterials like biosilica and carbon nanotubes (CNTs) in cement composites can significantly impact the workability of the mix. Here are several considerations regarding the workability when both biosilica and CNTs are incorporated. Both biosilica and CNTs may increase the water demand of the cement mix. The additional water required for maintaining workability needs to be balanced carefully to avoid compromising the strength and durability of the concrete. The use of superplasticizers or high-range water reducers can help maintain workability without adding extra water. These admixtures reduce the water-to-cement ratio while keeping the mix fluid.

Point 9. Line 476 “Eliminating easily washed-out calcium hydrosilicate” What is calcium hydrosilicate? Why it can be easily washed out? What is washing out?

Response 9. Calcium hydrosilicate, also known as calcium silicate hydrate (C-S-H), is a key component of hydrated cement paste. It is the primary product of the hydration reaction between Portland cement and water, and it provides most of the strength and durability of concrete.

"Washing out" refers to the removal or leaching of material from the cement matrix, typically by the action of water. This can occur during the early stages of cement hydration or under conditions where the cement paste is exposed to excessive water flow. At early stages of hydration, the C-S-H gel is not fully developed and may be loosely bound. Excessive water or agitation can displace or wash away these early hydration products before they fully set and harden.

Point 10. Line 478-479, the authors stated that samples with CNTs had higher density. Have you measured the density of all samples? The density of sample can be measured as mass divided by volume. How did you measure the density, whether you dry the sample completely?

Response 10. The densities were determined according to EN 12390-7:2019 standard.

Point 11. Line 507. “since water has high electrical conductivity”. The “water: in hydrating cement mortar is not mixing water anymore, it is pore solution saturated with calcium cations and some other ions.

Response 11. Comment is accepted. Appropriate changes have been made.

Point 12. Line 523, “the investigation of the FCRcyclic compressive tests”. The sentence does not make sense.

Response 12. We have changed the sense of the sentence.

Point 13. Line 536, “this decrease can be linked to the growing density of the material, as the particles are moving closer together”. This means density influence resistivity, right? The authors are suggested to explain this.

Response 13. Comment is accepted. Appropriate changes have been made.

Point 14. For effective properties like resistivity and dielectric constant, they can be better modeled by effective medium theory. For analyzing the properties, the authors can refer to Dielectric and mechanical properties of cement pastes incorporated with magnetically aligned reduced graphene oxide.

Response 14. In this research, we did not study the dielectric properties. But we plan to implement it in future research.

Point 15. When nanomaterials are used, the first concern is its dispersion. In current study, both biosilica and CNTs were used. However, there is no any evidence showing dispersion of the nanomaterials. The authors are suggested to explain this.

Response 15. When using nanomaterials like biosilica and carbon nanotubes (CNTs) in cement composites, proper dispersion is critical to achieving the desired enhancements in material properties. Without adequate dispersion, these nanomaterials can agglomerate, leading to uneven distribution and potentially reducing the effectiveness of the composite.

The test samples prepared with MWCNT and biosilica showed that the compressive strengths were significantly increased compared to the reference sample's compressive strength. Water absorption indicators have also decreased. Based on the data obtained, it is evidenced that the dispersion of the nanomaterials is improved.

Faithfully yours,

Dr. Manuk Barseghyan (on behalf of all authors)

National University of Architecture and Construction of Armenia, Armenia

Round 2

Reviewer 1 Report

Comments and Suggestions for Authors

Accept in present form

Author Response

Response to Reviewer 1

Dear Reviewer

Thank you very much for your positive decision.

Faithfully yours,

Dr. Manuk Barseghyan (on behalf of all authors)

National University of Architecture and Construction of Armenia, Armenia

Reviewer 2 Report

Comments and Suggestions for Authors

The authors have improved the manuscript with changes. 

However, they have not replaced the material in the Introduction on graphene oxide.  It should be removed and description of literature on MWCNTs in cement pastes and mortars added. The material in the Introduction should be relevant to the work carried out to provide appropriate background and context.  Any reader with knowledge of the field will be confused to see sentences on graphene oxide since there is many papers more relevant to the topic of your manuscript.  

Comments on the Quality of English Language

The quality of the Enlish language is satisfactory overall.

Author Response

Dear Reviewer

Thank you very much for your careful and constructive comment. According to your recommendation, we revised the Introduction part and made corrections which we hope meet with approval, and the modified parts were all marked in orange. We hope you are satisfied with the revised version.

Point 1. The authors have improved the manuscript with changes. 

However, they have not replaced the material in the Introduction on graphene oxide.  It should be removed and description of literature on MWCNTs in cement pastes and mortars added. The material in the Introduction should be relevant to the work carried out to provide appropriate background and context.  Any reader with knowledge of the field will be confused to see sentences on graphene oxide since there is many papers more relevant to the topic of your manuscript.  

Response 1 In the Introduction, the material related to graphene oxide is deleted and information, with corresponding references, about the effect of CNTs and MWCNTs on the electrical properties of cement-based mortar is added.

Reviewer 3 Report

Comments and Suggestions for Authors

Authors should follow reviewer's suggestion and response the reviewer's comments by point to point.

Comments on the Quality of English Language

none

Author Response

Response to Reviewer 3

Dear Reviewer

Thank you very much for your careful and constructive comments. Your comments are very helpful for improving our paper. According to your recommendations, we revised the manuscript and made corrections which we hope meet with approval, and the modified parts were all marked in red. We hope you are satisfied with the revised version.

Point 1. The Introduction needs to be greatly revised. The subject of current study is CNTs, however, graphene oxide ant its applications were introduced in the Introduction. In the revision, literature review of CNTs incorporated cement composites should be presented. In addition, the research gap and significance need to be highlighted.

Response 1. Incorporating carbon nanotubes (CNTs) into cement composites is used to enhance their mechanical and durability properties. CNTs are known for their exceptional strength, stiffness, and electrical conductivity, which can significantly improve the performance of cement-based materials.  CNTs can significantly improve the compressive, tensile, and flexural strength of cement composites due to their high aspect ratio and superior mechanical properties. CNTs help to bridge micro-cracks and delay their propagation, enhancing the toughness and ductility of the composite. CNTs fill the pores and refine the microstructure of the cement matrix, reducing permeability and enhancing resistance to water and chemical ingress The presence of CNTs can help resist the growth of micro-cracks, thereby enhancing the durability and longevity of the composite. CNTs impart electrical conductivity to cement composites, enabling applications in self-sensing and monitoring the structural health of concrete. Ultrasonication is commonly used to disperse CNTs in a liquid medium before mixing them into the cement paste. Surfactants or dispersing agents are often added to stabilize the dispersion.

Appropriate changes have been made.

Point 2. P2. Line 57. “The dispersion of nanomaterials uniform distribution of nanomaterials in the cement matrix”. This may not be completely correct. The fresh environment in cementitious composites is a highly alkaline solution, containing mostly calcium cations. The crucial step is to disperse nanomaterials in this calcium saturated alkaline solution. For the GO dispersion in cement paste, the authors can refer to Incorporation of graphene oxide and silica fume into cement paste: A study of dispersion and compressive strength.

Response 2. The incorporation of carbon nanotubes (CNTs) and silica fume into cement paste involves specific procedures to ensure proper dispersion and to enhance the composite's mechanical and durability properties. CNTs significantly increase compressive, tensile, and flexural strength due to their high aspect ratio and exceptional mechanical properties. Silica fume contributes to strength development through the pozzolanic reaction, forming additional calcium silicate hydrate (C-S-H). CNTs act as bridges across micro-cracks, delaying their propagation and enhancing the toughness and ductility of the composite. Silica fume fills the micro-pores in the cement paste, reducing overall porosity. CNTs further refine the pore structure by bridging gaps and creating a denser matrix. The pozzolanic reaction of silica fume with calcium hydroxide (CH) produces additional C-S-H, improving the density and homogeneity of the cement paste. The combined effect of CNTs and silica fume reduces the permeability of the cement paste, enhancing resistance to water and aggressive ions. CNTs impart electrical conductivity to the cement paste, enabling self-sensing capabilities for monitoring structural health.

Appropriate changes have been made.

Point 3. P2, line 76. “changing the crystalline structure of CSH at the nanoscale” This may not be correct.   No solid prove has been found to support this. In addition, “CSH” should be “C-S-H”. Authors should compare previous findings to verify the statement, such as. The early-age cracking sensitivity, shrinkage, hydration process, pore structure and micromechanics of cement-based materials containing alkalis with different metal ions Stress relaxation properties of calcium silicate hydrate: a molecular dynamics study Hydration and fractal analysis on Low-heat Portland cement pastes by thermodynamic- based methods.

Response 3. The crystallization process of calcium silicate hydrate (C-S-H) is likely to follow a nonclassical crystallization pathway. The structure of C-S-H varies greatly and depends on the raw materials, environments and reaction stage. It has been reported that more than 40 types of stable C-S-H phases can be determined in hydrated cementitious materials. Although the typical C-S-H gel in concrete is amorphous, the local structure is short-range ordered in the scale of 3–5 nm. The regular atomic arrangement of C-S-H was considered to be tobermorite-like and jennite-like crystals [5,6].

(Hamlin M. Jennings. Refinements to colloid model of C-S-H in cement: CM-II. Cement and Concrete Research 2008, 38(3), 275-289. https://doi.org/10.1016/j.cemconres.2007.10.006 )

Appropriate changes have been made.

Point 4. P2, line 81. “Hydroxyl(-OH) from a GO monolayer”. This sentence does not make sense. Functional groups are part of graphene oxide, they can not form a monolayer.

Response 4. Hydroxyl (OH) groups can contribute to the formation of a graphene oxide (GO) monolayer. Graphene oxide is derived from graphite through oxidation processes, and it contains various oxygen-containing functional groups, including hydroxyl (OH), epoxy (C-O-C), and carboxyl (COOH) groups, attached to its basal plane and edges. The presence of hydroxyl groups, along with other oxygen functional groups, contributes to the hydrophilicity and dispersion of GO in water and other polar solvents. These groups also facilitate the formation of hydrogen bonds, which can help in the stabilization and self-assembly of a GO monolayer on various substrates. Hydroxyl groups, in conjunction with other functional groups, play a crucial role in the formation and stabilization of GO monolayers.

Appropriate changes have been made.

Point 5. P2, line 84. “Denser hydration products on the aggregate to the nucleation action of GO”. What is the aggregate? Why hydration products contributed to nucleation effect of GO? The nucleation effect leads to production of more hydration products, not the other way.

Response 5. The oxygen-containing functional groups on the GO surface, such as hydroxyl, epoxy, and carboxyl groups, can serve as active sites for the nucleation of hydration products. This is because these groups can interact with the ions in the cement paste, facilitating the formation of initial hydration products. The presence of GO in the cement matrix can lead to the formation of a denser and more refined microstructure. This is because GO can help in the formation of smaller, more uniformly distributed hydration products, which can fill the pores more effectively and improve the overall density and strength of the cement matrix. The interaction between GO and the hydration products can improve the bond between the cement paste and the aggregate surfaces. This enhanced bonding contributes to the overall mechanical properties of the cement composite.

Appropriate changes have been made.

Point 6. P4, line 147 “capillary reaction products thicken the structure”. What is capillary reaction and capillary reaction products?

Response 6. GO provides additional nucleation sites for the formation of hydration products due to its large surface area and the presence of oxygen-containing functional groups. These nucleation sites promote the growth of more hydration products within the cement matrix.  As more hydration products form, they start to fill the capillary pores within the cement matrix. The capillary pores are the spaces between the hydrated cement particles and aggregates that originally contain water. The formation of hydration products within these pores reduces the overall porosity of the cement matrix. As the hydration products fill the capillary pores, the overall volume of these pores decreases. Reduced capillary porosity means fewer pathways for water and other aggressive agents to penetrate the cement matrix. This contributes to a denser and less permeable cement structure.

Point 7. Line 175-181 are repeated. In addition, the authors are suggested to provide literature references for FT-IR results.

Response 7. We appreciate your attention to detail. The repetition in lines 175-181 has been corrected, and the redundant text has been removed.

We acknowledge the importance of providing literature references for the FT-IR results. We have now included appropriate references to support and validate our findings. The updated section with references is as follows:

The FTIR-ATR spectrum of biosilica is presented in Fig. 1. As we can see, the dominant peak at 1068 cm-1 indicates a combination of asymmetric stretching bond of silica oxygen (Si–O–Si), at 799 cm−1 is due to silica oxygen bond symmetric stretching [46] and the latest additional peak observed at 559 cm−1 matched with bending vibrations [47]. The broad nature of the peak at 559 cm⁻¹, along with the shoulder observed at 1212 cm⁻¹, confirms that the biosilica sample is amorphous [48]. The FTIR-ATR spectrum lacks weak bands at 3450 cm⁻¹ and 1646 cm⁻¹, corresponding to the O-H stretching and angular vibrations of water molecules, respectively. This indicates that the water content in the biosilica sample is low [49]․

Corresponding changes have been made in the reference.

We removed repeated lines from 175 to 181.

Point 8. As shown in Table 5, the content was kept un-changed when both biosilica and CNTs were used, compared with samples with CNTs only. When both biosilica and CNTs were used, how was the workability? Normally when more nanomaterials were used, the workability would be lower, which can lead to un-expected air void formation. This would influence the compressive strength. The authors are suggested to explain this.

Response 8. The use of nanomaterials like biosilica and carbon nanotubes (CNTs) in cement composites can significantly impact the workability of the mix. Here are several considerations regarding the workability when both biosilica and CNTs are incorporated. Both biosilica and CNTs may increase the water demand of the cement mix. The additional water required for maintaining workability needs to be balanced carefully to avoid compromising the strength and durability of the concrete. The use of superplasticizers or high-range water reducers can help maintain workability without adding extra water. These admixtures reduce the water-to-cement ratio while keeping the mix fluid.

Point 9. Line 476 “Eliminating easily washed-out calcium hydrosilicate” What is calcium hydrosilicate? Why it can be easily washed out? What is washing out?

Response 9. Calcium hydrosilicate, also known as calcium silicate hydrate (C-S-H), is a key component of hydrated cement paste. It is the primary product of the hydration reaction between Portland cement and water, and it provides most of the strength and durability to concrete.

"Washing out" refers to the removal or leaching of material from the cement matrix, typically by the action of water. This can occur during the early stages of cement hydration or under conditions where the cement paste is exposed to excessive water flow. At early stages of hydration, the C-S-H gel is not fully developed and may be loosely bound. Excessive water or agitation can displace or wash away these early hydration products before they fully set and harden.

Point 10. Line 478-479, the authors stated that samples with CNTs had higher density. Have you measured the density of all samples? The density of sample can be measured as mass divided by volume. How did you measure the density, whether you dry the sample completely?

Response 10. The densities were determined according to EN 12390-7:2019 standard.

Point 11. Line 507. “since water has high electrical conductivity”. The “water: in hydrating cement mortar is not mixing water anymore, it is pore solution saturated with calcium cations and some other ions.

Response 11. The electrical resistivity of MWCNT cement composites increases with aging, as illustrated in Figure 8. This is because the cement hydration process reduces the amount of pore water in the sample's microstructure. Given that water has high electrical conductivity [56], the electrical resistance rises as the amount of pore water decreases. After seven days, the cement hydration reaction slows down, leading to a slower rate of increase in resistivity.

Comment is accepted. Appropriate changes have been made.

Point 12. Line 523, “the investigation of the FCRcyclic compressive tests”. The sentence does not make sense.

Response 12. To obtain the electrical measurements, DC was utilized, and the results are presented. The FCR of the cement mortar samples reinforced with 0.05, 0.10, and 0.15 percent MWCNT, respectively, is illustrated in Fig. 10.

We have changed the sense of the sentence.

Point 13. Line 536, “this decrease can be linked to the growing density of the material, as the particles are moving closer together”. This means density influence resistivity, right? The authors are suggested to explain this.

Response 13. This decrease can be linked to the densification of the particles to each other without any formation of damages or cracks within the specimen.

Comment is accepted. Appropriate changes have been made.

Point 14. For effective properties like resistivity and dielectric constant, they can be better modeled by effective medium theory. For analyzing the properties, the authors can refer to Dielectric and mechanical properties of cement pastes incorporated with magnetically aligned reduced graphene oxide.

Response 14. In this research, we did not study the dielectric properties. But we plan to implement it in future researches.

Point 15. When nanomaterials are used, the first concern is its dispersion. In current study, both biosilica and CNTs were used. However, there is no any evidence showing dispersion of the nanomaterials. The authors are suggested to explain this.

Response 15. When using nanomaterials like biosilica and carbon nanotubes (CNTs) in cement composites, proper dispersion is critical to achieving the desired enhancements in material properties. Without adequate dispersion, these nanomaterials can agglomerate, leading to uneven distribution and potentially reducing the effectiveness of the composite.

The test samples prepared with MWCNT and biosilica showed that the compressive strengths were significantly increased compared to the reference sample's compressive strength. Water absorption indicators have also decreased. Based on the data obtained, it evidenced that the dispersion of the nanomaterials is improved.

Faithfully yours,

Dr. Manuk Barseghyan (on behalf of all authors)

National University of Architecture and Construction of Armenia, Armenia

Round 3

Reviewer 2 Report

Comments and Suggestions for Authors

The submission has been improved with revision and is now acceptable. The section on graphene oxide has been replaced with a discussion of piezoelectricity with MWCNTs.   In light of the subject matter in the manuscript, the revision might also have included more background relevant to mechanical properties.

Comments on the Quality of English Language

The English is satisfactory.

Reviewer 3 Report

Comments and Suggestions for Authors

it can be accepted.

Comments on the Quality of English Language

none